ecology, evolution

developmental plasticity, $G \times E \times E$, intraspecific variation, temperature, nutrition, multidimensional plasticity

**Author for correspondence:**
Nadja Verspagen
e-mail: nadja.verspagen@helsinki.fi

# Multidimensional plasticity in the Glanville fritillary butterfly: larval performance is temperature, host and family specific

Nadja Verspagen[1,2,3], Suvi Ikonen[2,3], Marjo Saastamoinen[1,2] and Erik van Bergen[1,2]

[1]Helsinki Institute of Life Science, University of Helsinki, Finland
[2]Research Centre of Ecological Change, Faculty of Biological and Environmental Sciences, University of Helsinki, Finland
[3]Lammi Biological Station, University of Helsinki, Finland

NV, 0000-0002-3526-3965; MS, 0000-0001-7009-2527; EvB, 0000-0002-9648-9837

Variation in environmental conditions during development can lead to changes in life-history traits with long-lasting effects. Here, we study how variation in temperature and host plant (i.e. the consequences of potential maternal oviposition choices) affects a suite of life-history traits in pre-diapause larvae of the Glanville fritillary butterfly. We focus on offspring survival, larval growth rates and relative fat reserves, and pay specific attention to intraspecific variation in the responses ($G \times E \times E$). Globally, thermal performance and survival curves varied between diets of two host plants, suggesting that host modifies the temperature impact, or vice versa. Additionally, we show that the relative fat content has a host-dependent, discontinuous response to developmental temperature. This implies that a potential switch in resource allocation, from more investment in growth at lower temperatures to storage at higher temperatures, is dependent on the larval diet. Interestingly, a large proportion of the variance in larval performance is explained by differences among families, or interactions with this variable. Finally, we demonstrate that these family-specific responses to the host plant remain largely consistent across thermal environments. Together, the results of our study underscore the importance of paying attention to intraspecific trait variation in the field of evolutionary ecology.

## 1. Introduction

Species can cope with environmental change by avoiding stressful conditions, by producing phenotypes better adjusted to the new environmental conditions through plasticity, or by adapting to the novel conditions through evolutionary change [1,2]. Even though the avoidance of environmental stress is an effective strategy (e.g. through tracking favourable conditions by expanding to higher latitudes or altitudes [3] or by selecting more suitable microhabitats), it is often limited by factors such as the distribution of resources, the structure of the landscape and/or the dispersal ability of the species. Moreover, when environmental changes are rapid, adaptive evolution may not occur fast enough. In those cases, plasticity can enable species to persist under the novel conditions, allowing more time for mutations to arise and selection to occur [4,5]. Assessing a species's ability to respond plastically to environmental change and evaluating its performance when exposed to conditions that are beyond or at the limit of the normal range could therefore shed light on whether organisms will be able to persist future conditions.

Developmental plasticity is defined as the process through which external conditions, such as nutrition and temperature, can influence developmental trajectories and lead to irreversible changes in the adult phenotype [1]. This phenomenon is ubiquitous in nature, especially among taxa that have sessile lifestyles [6–8]. The environmental regulation of development has been studied extensively using insects, whose pre-adult stages are often immobile and thus must cope with local environmental conditions. In general, when exposed to higher temperatures, insect larvae tend to grow faster [9,10] and the body size of the emerging adults is smaller [9,11,12], which might alter performance later in life. Likewise, nutrition is known to regulate development in insects through nutrient balance [13–15] and/or the concentration of secondary metabolites in the diet [16].

When assessing responses to changes in environmental conditions, it is important to recognize that the environmental factors that affect the phenotype typically occur simultaneously and interactively [17]. Hence, plastic responses to one type of environmental stress may be dependent on the state of another external factor. Such non-additive multidimensional plasticity, in response to combinations of thermal and nutritional environments, has been demonstrated in moths [14], butterflies [18] and fruit flies [19]. For example, Singh *et al.* [18] showed that poor host plant quality mainly influenced development at intermediate temperatures in the tropical butterfly *Bicyclus anynana*. Moreover, significant genetic variation for (multidimensional) plasticity is known to exist in both natural and laboratory populations [20–22]. This intraspecific variation in the ability to respond to an environmental cue (G × E), or combinations of cues (G × E × E), is hypothesized to be beneficial in the context of climate change since it facilitates the evolution of wider ranges of environmental tolerance [23,24].

In this study, we focus on environmentally induced variation in a suite of life-history traits in the Glanville fritillary butterfly (*Melitaea cinxia*). The species occurs at its northern range margin on the Åland archipelago (southwest Finland), where it inhabits a highly fragmented network of habitat patches that are defined by the presence of at least one of two available host plant species: *Plantago lanceolata* and *Veronica spicata*, hereafter referred to as *Plantago* and *Veronica*, respectively [25]. Adult females produce large egg clutches, and the selection of suitable oviposition sites is known to be a hierarchical process [26,27]. In the field, gravid females of the Glanville fritillary appear to first choose habitats that are hot, dry and sunny [28,29]. Host plant discrimination, with individuals typically preferring one host species over the other, occurs subsequently [30,31]. Therefore, selective mothers can influence the developmental trajectories of their offspring through oviposition site selection, which in turn may affect offspring performance and fitness [32].

Using a full-factorial split-brood design, we explore the consequences of these maternal oviposition choices for the pre-diapause larvae of *Melitaea cinxia*. We aim to research the (combined) effects of developmental temperature and host plant on pre-diapause larvae of this species. We measure the survival, larval growth rates and relative fat content of offspring reared at four different temperatures and on two different host plants, and pay attention to intraspecific variation in the responses by using individuals from different genetic backgrounds (i.e. families). As shown in other insects, we expect a large positive effect of developmental

temperature on growth rate. Furthermore, in the scenario of additive multidimensional plasticity, we expect larvae to grow faster and have higher survival on *Veronica* within each thermal environment, as this has previously been demonstrated under laboratory conditions [16,33]. Individuals that develop fast, and thus will be diapausing for longer, are predicted to allocate relatively more resources to fat storage, since fat is thought to be the primary fuel for overwintering and post-winter activities in insects [34,35]. Finally, given that the natural habitat of this species is heterogeneous, fragmented and highly variable we expect family-specific responses to the environmental factors (G × E, G × E × E) to be important determinants of the phenotype.

## 2. Methods

### (a) Study system

*Melitaea cinxia* is a univoltine species and on the Åland islands adults emerge from their pupae in June, after which females lay several clutches of 100–200 eggs [36]. The sessile pre-diapause larvae hatch in late June and early July and live gregariously on the host plant of their mothers' choice. In the beginning of autumn the larvae spin a communal web in which they diapause until spring. Overwinter survival is impacted by, among other factors, body size, with larger larvae having a higher chance to survive [37]. After diapause, larvae become solitary and can move over longer distances in search of resources and/or suitable microhabitats [38]. The laboratory population of *M. cinxia* used in this study was established in 2015 from 136 post-diapause larvae (consisting of 105 unique families) collected from 34 habitat patches across the large network of habitat patches on the Åland islands. Laboratory stocks of *M. cinxia* are maintained on *Plantago* host plants in climate-controlled chambers set to 28°C during the day and 15°C during the night (28 : 15°C), and a 12 L : 12 D cycle. These settings mimic the microclimatic conditions that the larvae are exposed to during summer in Åland at the ground level. Diapause is obligatory in this species and diapausing larvae are stored until the next spring in cooling chambers set to 5°C and complete darkness [37].

### (b) Experimental design

In the spring of 2019, diapausing larvae of the laboratory stock were removed from the cooling chamber and placed under standard rearing conditions (28 : 15°C and a 12 L : 12 D cycle) where the increase in temperature, light and moisture stimulates them to recommence development. This parental generation was reared to adulthood on *Plantago* host plants in small transparent plastic containers, and mated with an unrelated individual. Subsequently, gravid females were provided with a single *Plantago* plant for oviposition, and the host plant was checked daily for newly produced egg clutches. Clutches were carefully removed, placed in individual petri-dishes, and transferred to a climate-controlled cabinet set to standard rearing conditions (28 : 15°C and a 12 L : 12 D cycle).

Egg clutches of fourteen females were split across two host plant treatments (*Veronica* and *Plantago*) 3–5 days after oviposition to ensure the utilization of a single host plant species throughout development. When approximately 90% of the larvae within each group transitioned from the first to the second instar, larval groups were further divided into cohorts of 15 siblings. The time between the first larva transitioning to the second instar and brood splitting was less than one day for the majority of the families. These full-sib cohorts were randomly divided over four climate-controlled chambers (with 28°C, 30°C,

32°C or 34°C during the day, 15°C at night, and a 12 L : 12 D cycle), using a Sanyo MLR-350 for the 32°C treatments and a Sanyo MLR-351 for the others. Starting from the standard rearing conditions (28 : 15°C), these 2°C increments in day temperature were chosen to reflect the microclimatic conditions of pre-diapause larvae in habitats that are relatively hot and sunny. This procedure resulted in a full-factorial split-brood experimental design with fourteen genetic backgrounds (i.e. families), two host plants and four developmental temperatures ($n = 1680$). Throughout the experiment, larvae were inspected every morning and fresh leaves were provided to ensure *ad libitum* feeding conditions. For five families, offspring from a second egg clutch (from the same parents) were used to complete all experimental treatments. A schematic of the experimental design is given in electronic supplementary material, figure S1. For further information on the background of larvae used in the experiment, see electronic supplementary material, table S1.

## (c) Life-history traits

We studied environmentally induced variation in a suite of developmental life-history traits and focused on offspring survival, larval growth rates, and the relative amount of fat reserves accumulated during larval development. To assess offspring survival, the larvae within each cohort were counted every fourth day, and on these days the entire cohort was weighed to the nearest 0.01 mg (Mettler Toledo XS105 DualRange) to trace overall mass gain during larval development. The total mass of the cohort was divided by the number of surviving individuals to obtain the mean larval mass for each time-point. This procedure was continued until the first individual of the cohort entered the diapause stage, which can be recognized by a change in body colour (from pale-brown to black), an increase in larval body hair density and the presence of red eyes. From this date forward, individual data were collected by recording the day of entering diapause and the body mass of each diapausing larvae. Subsequently, larvae were frozen to −80°C, and stored in eppendorf tubes until further processing.

The individual growth rates were calculated according to the formula

$$\text{growth rate} = \frac{\ln(\text{diapause mass}) - \ln(\text{2nd instar mass})}{\text{development time}},$$

where 2nd instar mass (i.e. mass at the start of the experiment) was estimated by dividing the mass of the entire cohort by the number of individuals, and development time was computed as the time between the start of the experiment and the day the individual entered diapause [39].

Relative fat content at diapause was determined for seven randomly chosen individuals per cohort ($n = 784$). These larvae were dried to constant mass (60°C for 24 h) and weighed to the nearest 0.01 mg, yielding initial dry mass. Triglyceride and free fatty acids were extracted by incubating the dried body at room temperature in 1 : 2 (v/v) methanol:dichloromethane for 72 h, followed by drying and re-weighing, yielding fat-free dry mass [40]. The relative fat content was calculated according to the formula

$$\text{relative fat content} = \frac{(\text{initial dry mass} - \text{fat-free dry mass})}{\text{initial dry mass}}.$$

## (d) Statistical analyses

Interval-censored survival curves were fitted using the *survival* package [41] and plotted using the *survminer* package [42]. For each time interval, larvae were categorized as dead or alive, and diapausing larvae were censored because they were no longer at risk of pre-diapause death. Log-rank tests were performed to determine the influence of temperature and host plant on survival using the *interval* package [43]. To confirm the obtained results, a generalized linear mixed effects model (GLMM) was fitted to the endpoint survival data (state of the larvae (dead or alive) at the end of the experiment), using a binomial distribution and the *lme4* package [44].

A linear model was fitted to estimate the effect of temperature and host on the mean amount of body mass gained during larval development. Mean larval mass, computed as stated above, was log-transformed to improve normality. The day of the experiment, temperature and host plant (and all interactions) were included as fixed factors in the full model. Two additional linear models were fitted to estimate the (fixed) effects of family, temperature and host (and all interactions) on individual growth rate and relative fat content. For all models described above, step-wise model selection based on AIC values was performed using the *step()* function. Post hoc pairwise comparisons (Tukey's HSD; $\alpha = 0.05$) were performed using the *emmeans* package [45].

Intraspecific variation in the responses to the host plant is explored by extracting the slope of a linear model—with individual growth rate as dependent and host plant as independent variable—for each family and within each thermal environment. These slopes describe both the magnitude and the direction of the response to the host plant [20]. Using Pearson correlations we test whether host-induced responses (i.e. the slopes) are family-specific and consistent across thermal environments. All statistical analyses were performed in R [46].

## 3. Results

## (a) Pre-diapause survival and clutch mass

Probability of survival was generally high ($n = 1559$; 93%) but dropped considerably for larvae with long development times (i.e. those that enter diapause in the upper percentiles of the distribution of development times; figure 1). The probability of survival within each host plant was not affected by temperature (asymptotic log-rank two-sample *t*-test, $p = 0.3968$ for individuals reared on *Plantago*, and $p = 0.8678$ for individuals reared on *Veronica*). However, within each temperature, survival was significantly lower for larvae that were reared on *Plantago*, but only for the two highest temperatures (*p*-values given in figure 1). These results were confirmed by the GLMM using the endpoint data (see electronic supplementary material, table S2).

The average fresh mass of the larvae increased over time, and the rate of this increase was affected by the termal environment and the host plant. The effect of temperature on mean larval mass increased with time (time : temperature, $F = 10.5182$, $p < 0.001$), with larvae reared at 28°C growing slower overall, and being significantly smaller than those reared at higher temperatures from day 8 onward (figure 2a). The mean larval mass of cohorts reared on *Veronica* increased faster over time compared to those reared on *Plantago* (time : plant, $F = 3.9190$, $p = 0.0089$). Cohorts using *Veronica* were smaller than those using *Plantago* at the start of the experiment (day 0; figure 2a) but larger at the final time point (day 16).

## (b) Individual growth rates and allocation to fat reserves

For both life-history traits (growth rate and fat content), we found that all main effects and all interaction terms were

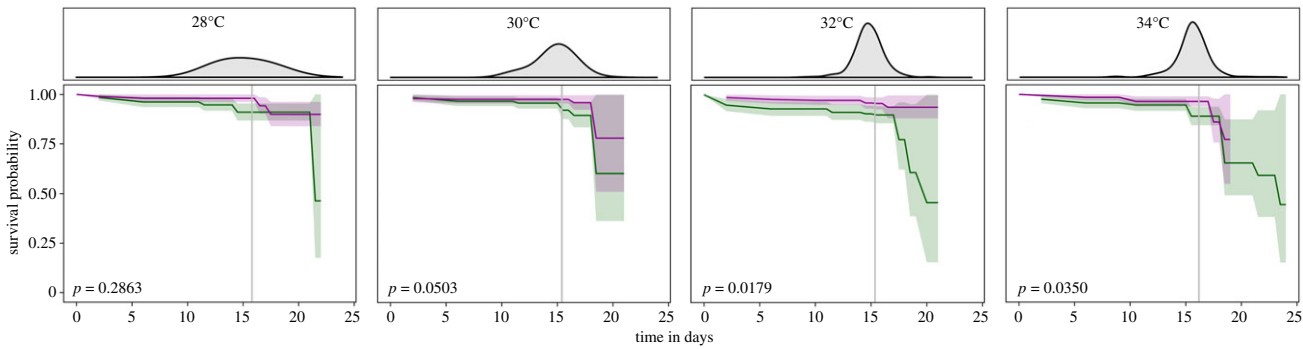

**Figure 1.** Kaplan–Meier survival probability over time for larvae reared on *Plantago* (green) and *Veronica* (purple), at four different day temperatures. Shaded area represents the 95% confidence interval. Note that larvae were censored at the time point of entering diapause (because they are no longer 'at risk'), leading to smaller sample sizes and larger confidence intervals towards the end of survival assessments. Grey lines show the mean day of diapause, and the distribution of diapausing day is given in the upper panels. The probability of survival is not affected by temperature, but, at the two highest day temperatures, survival is significantly lower for larvae that were reared on *Plantago* (*p*-values given in the figure). (Online version in colour.)

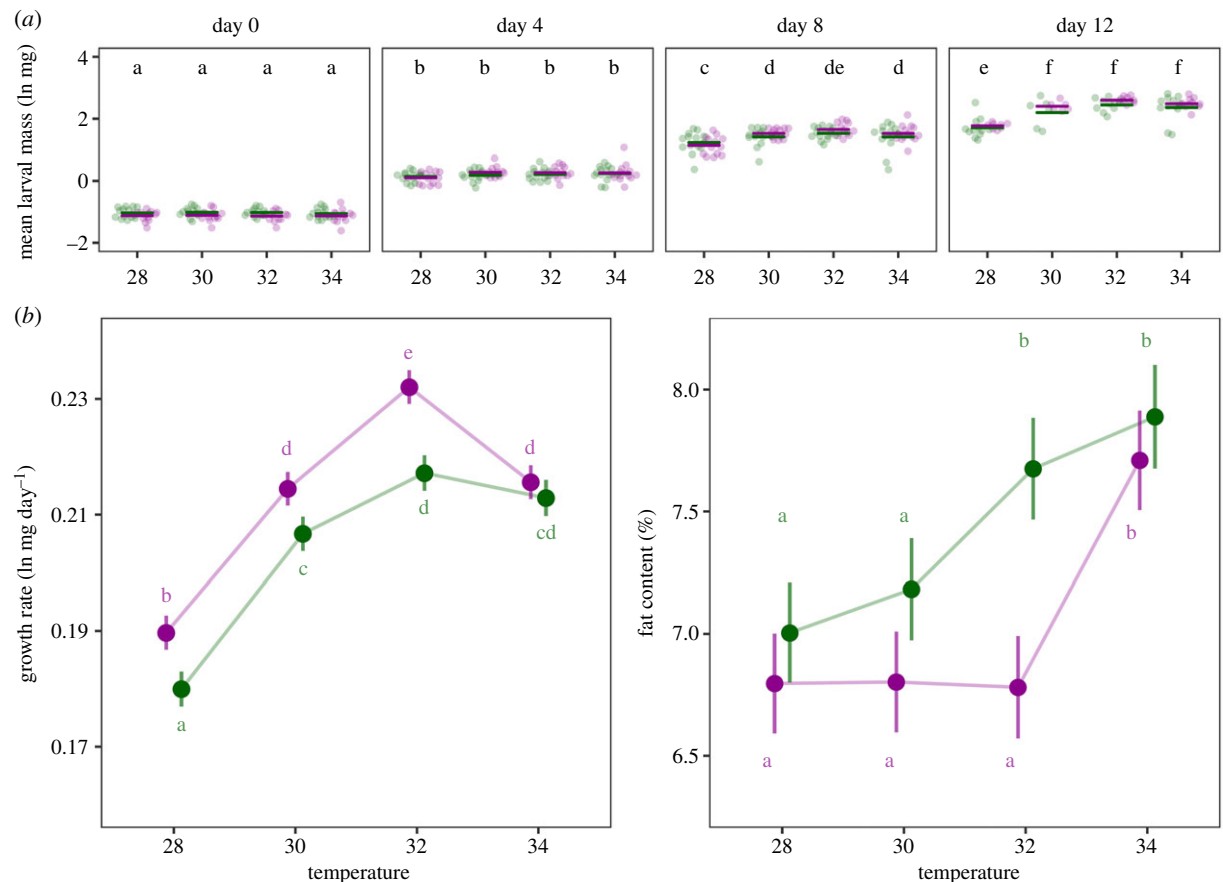

**Figure 2.** Environmentally induced variation in life-history traits. (*a*) Dots depict the mean larval mass, log-transformed and corrected for number of individuals, in each thermal environment (*x*-axis), for *Plantago* (green) and *Veronica* (purple), on four assessment days (from left to right: day 0 [i.e. 2nd instar mass], day 4, day 8 and day 12). Significant differences between thermal treatments (Tukey's HSD, $\alpha = 0.05$) are indicated by different letters. Details of the statistical test can be found in electronic supplementary material, table S3. (*b*) Model-estimated marginal means for the individual growth rates (left; $R^2 = 0.5964$) and the relative fat content (right; $R^2 = 0.4992$). Error bars represent 95% confidence intervals and significant differences between groups (Tukey's HSD, $\alpha = 0.05$), averaged over the families, are indicated by different letters. Details of statistical tests can be found in electronic supplementary material, tables S3 and S4. (Online version in colour.)

statistically significant (see electronic supplementary material, tables S3 and S4). Averaged over the families, model-estimated marginal means for the individual growth rates revealed that individuals achieve higher growth rates on *Veronica*, except for those reared at 34°C (figure 2*b*; electronic supplementary material, table S4C). Growth rate increased with temperature until a maximum at 32°C.

At an even higher temperature of 34°C growth rate dropped significantly compared to that at 32°C for larvae fed with *Veronica* (pairwise comparison: $p < 0.001$). In contrast, the growth rates of larvae reared at 34°C on *Plantago* were not significantly different from those of individuals reared at 32°C (pairwise comparison: $p = 0.5213$). This decrease in growth rate at 34°C was mainly caused by an increase in

development time rather than a decrease in body mass (electronic supplementary material, figure S2).

The relative fat content showed a discontinuous change to the temperature gradient on both hosts (figure 2b; electronic supplementary material, table S5C). For individuals reared on *Plantago*, development at the two higher temperatures resulted in significantly higher relative amounts of fat reserves. The thermal threshold at which change in relative fat content occurs was higher for individuals reared on *Veronica*, where only development at the highest temperature lead to an increase in relative fat content. As a result of the difference in threshold, we only observed a significant effect of host plant at 32°C (pairwise comparison: $p < 0.001$), with larvae using *Plantago* having a higher relative fat content on average.

## (c) Family-specific responses to the host plant

Our results demonstrate that intraspecific variation for multidimensional phenotypic plasticity (G × E × E) is large in this system. For both life-history traits, but especially for the individual growth rates, the (interactive) effects of environmental cues were highly dissimilar across families. About 12% of the total phenotypic variance ($V_P$) in individual growth rates was explained by the interaction between the family and the host plant (family:host plant, $F = 32.2507$, $p < 0.001$; see electronic supplementary material, table S4B). In other words, family-specific responses to the host were an important determinant of the phenotype. Indeed, some families used in the experiment achieved the highest growth rates on *Veronica* while individuals from other families grew consistently faster on *Plantago* (figure 3). These family-specific reaction norm slopes were positively correlated across thermal environments (Pearson's $r = 0.4$–0.8; electronic supplementary material, figure S3). Moreover, using *Plantago* as a host plant resulted in higher variance in larval growth rates across families (and not within families; electronic supplementary material, figure S4).

## 4. Discussion

Using a full-factorial design, with fourteen genetic backgrounds (families), four developmental temperatures and two host plant species, we explored the relative contributions of different sources of phenotypic variance across a suite of life-history traits in the Glanville fritillary butterfly. We start this section by describing the general patterns observed in our data, and then discuss how the developmental trajectories of pre-diapause larvae could be influenced by maternal oviposition site selection in the wild. Subsequently, we go into the variation in environmental responses observed among families in our study, and discuss how this genetic variation for (multidimensional) plasticity may impact the population's ability to persist environmental heterogeneity.

In ectotherms, temperature can affect developmental processes directly through changes in chemical reaction kinematics and the physical properties of membranes [47,48], which in turn can impact organismal performance and fitness. Some developing individuals are able to change their thermal environment, for example, by relocating to warmer microhabitats. Alternatively, when immature life-stages are largely immobile, such as in the case of the Glanville fritillary butterfly, the optimal thermal environment for development can be realized through selective oviposition choices of the female. As is true for many butterfly species, Glanville fritillary mothers could regulate the thermal environment of their offspring by preferring sunny or shady environments for oviposition [28,29]. Averaged over the families, our data showed a clear initial increase in growth rate with increasing temperature, with an optimum around 32°C. At higher temperatures the growth rate decreased for larvae reared on *Veronica* and stabilized for those reared on *Plantago*.

It is worth noting that evaluating the full consequences of our results would require assessments of how variation in pre-diapause performance translates into variation in reproductive success. Nevertheless, overwinter survival is known to be reduced in small-bodied individuals [37] and, since effects of reduced growth typically carry over to later life-stages [49,50], larval growth rates are commonly used as a proxy for offspring quality in Lepidoptera [51,52]. Even though strategies to compensate for a lower growth rate during pre-diapause development exist, these tend to come with a cost of delayed emergence, reduced fecundity and/or adult lifespan [37,53]. Thus, maternal preferences to oviposit in sunny habitats, thereby increasing the developmental temperatures of their offspring, could intuitively be considered adaptive in the Glanville fritillary butterfly. However, even though the average ambient day temperatures on Åland are well below 32°C during the end of summer, when pre-diapause larvae are developing (electronic supplementary material, figure S5), sunshine creates thermal stratification which can cause the temperatures close to the ground to rise to be between 12°C and 20°C above ambient temperature [54,55]. This suggests that the maternal preference for sunny habitat could in fact be maladaptive when the ambient day temperatures on Åland are above 20°C (see also [29]). In this scenario developmental temperatures in sunny microclimates may rise well above the observed optimal temperature of 32°C for larval growth, and potentially even exceed the thermal tolerance limits of the larvae. It was recently shown that the summer of 2018 was an anomaly in terms of precipitation, temperature and vegetation productivity across the habitats of the Glanville fritillary butterfly in Åland, and that this extreme climatic event was associated with a 10-fold demographic decline of the metapopulation [56]. Though this dramatic decline has been attributed to severe water deficits during May and July, the record-breaking temperatures observed for July (above 25°C; electronic supplementary material, figure S5) could have exceeded the thermal tolerance limits of pre-diapause larvae developing in sunny environments.

In addition to an effect of temperature on growth rates, our data also reveal a general trend in the relative fat content of the larvae. This physiological trait is important for butterfly life-history [57] and was quantified for the first time in this species. Relative fat content increases significantly between 30°C and 32°C for larvae reared on *Plantago*, and between 32°C and 34°C for larvae reared on *Veronica*. Previous research has shown similar increases in relative fat content with increasing temperature in other insects [e.g. 58–60], while other studies have described the opposite pattern (e.g. [40,61]). Our hypothesis, stating that individuals predicted to spend more time in diapause (i.e. those with shorter development times), accumulate more fat during development, was therefore incorrect. In fact, the cohorts with the largest relative fat reserves also demonstrated the

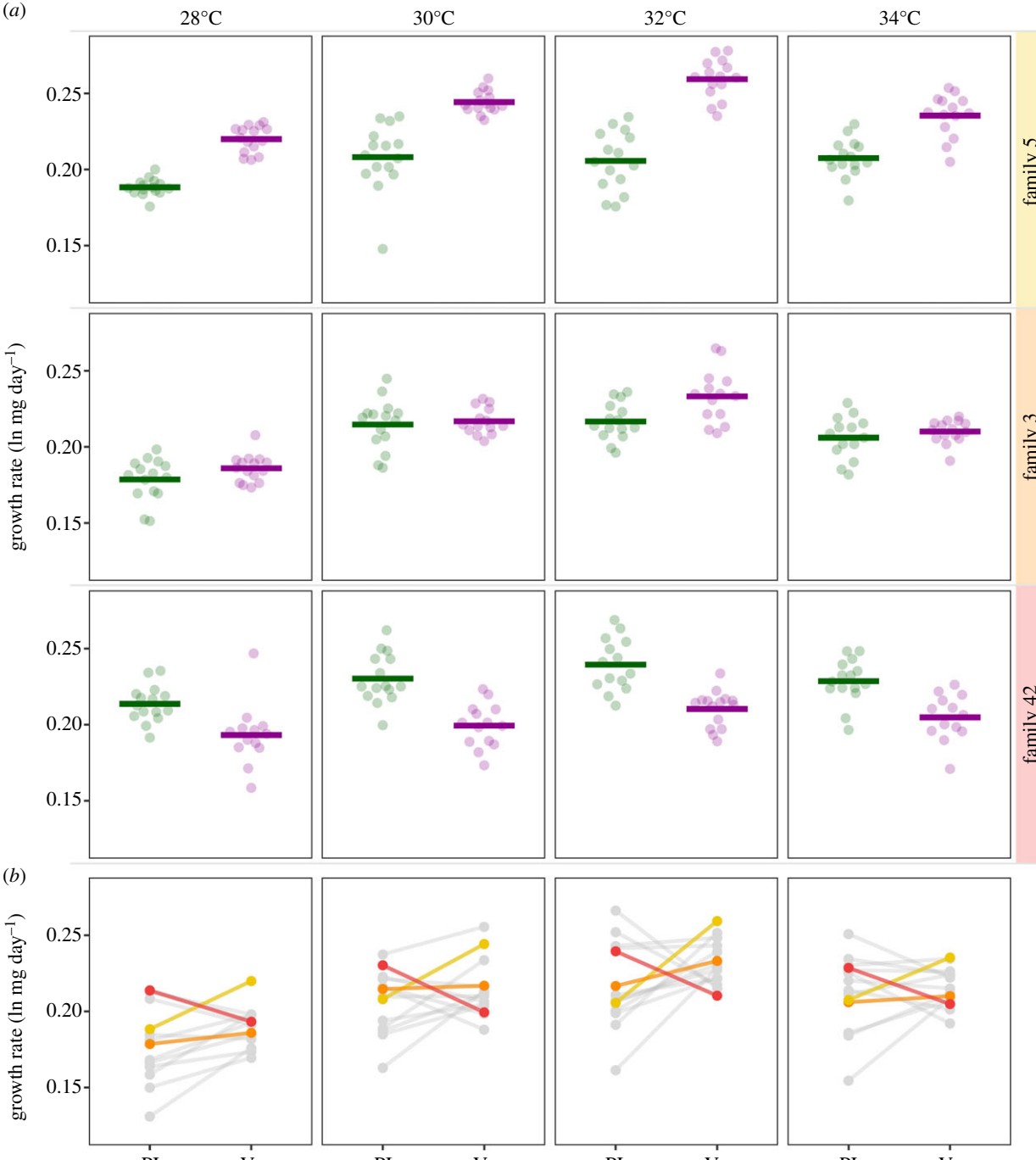

**Figure 3.** Host-induced responses in growth rates vary across families, but are consistent across thermal environments. (*a*) Panels demonstrate the individual growth rates on *Plantago* and *Veronica* of three representative families. Siblings of some families consistently achieve higher growth rates on *Veronica* (upper panels; yellow), while individuals from other families demonstrate an equal performance on each host plant (central panels; orange) or even grow faster on *Plantago* (lower panels; red). (*b*) Norms of reaction to the host plant for all families included in the experiment, coloured families correspond to those given in panel (*a*). The reaction norm slopes, describing both the magnitude and the direction of the response, are family-specific and correlate strongly across thermal environments (for details see electronic supplementary material, figure S3A). Using *Plantago* leads to higher between-family variance in growth rates (for details, see electronic supplementary material, figure S3B). (Online version in colour.)

longest development times and thus shortest time in diapause (i.e. those reared at 34°C). Instead, variation in relative fat content seemed to be associated with the relative investment in growth. Individuals reared at 32°C allocated more resources to their fat reserves when they used *Plantago* instead of *Veronica*. At this temperature, we also observed the largest difference in host-specific growth rates, with individuals using *Plantago* demonstrating reduced investment in growth compared to their siblings reared on *Veronica*. This suggests that individuals of this species compromise growth for increased investment in fat storage at higher

temperatures, and that the thermal threshold for this switch in life-history strategies is influenced by the larval diet.

Within the preferred sunny habitats, mothers select one of two available host plants, which may differ in their suitability for larval development. Overall, and in concordance with our hypothesis and earlier reports [16,33], we found that prediapause larvae perform better on *Veronica* than on *Plantago*. Though survival in the first instars was high on both plant species, the probability of survival to diapause was higher for larvae that were reared on *Veronica*, particularly in the two hottest environments. In addition, individuals on this

host achieved higher average growth rates in most thermal environments. Finally, the larval performances of the families used in this study were very uniform when using *Veronica*, which was in stark contrast to the high variance in growth rates observed among families on *Plantago*. Overall, the results of our laboratory study are in line with results from large-scale modelling efforts using the long-term demographic data of the Glanville fritillary metapopulation. These studies have demonstrated that the presence of *Veronica* in the habitat patch is associated with higher overall abundances [62], and that population growth rates of *M. cinxia* are reduced when temperatures during the summer are high [63].

Since females can maximize their fitness by laying their eggs on host plants on which the performance of their offspring is maximized (preference–performance hypothesis), Glanville fritillary mothers are predicted to prefer *Veronica* when both hosts are available. Interestingly, it has been shown that females of this species do not necessarily prefer the host plant that is most abundant in their local environment, but that this preference depends on which host is more abundant at a larger regional scale [30,64,65]. This local adaptation is attributed to the spatial distribution of the two hosts in the field, with *Plantago* being omnipresent and *Veronica* mainly occurring in habitat patches in the northwestern part of the archipelago. Females from regions where *Veronica* is an abundant and therefore reliable host plant, were observed to prefer *Veronica* when offered a binary choice, while butterflies in regions where *Veronica* is less reliable preferred to oviposit on *Plantago* [30,65].

In addition to the more general patterns described above, we found significant genetic variation for (multidimensional) plasticity in this system. In other words, the phenotypic responses to the (combination of) environmental variables were highly dissimilar across families (i.e. significant G × E and G × E × E interactions). For example, the interaction between the family and the host plant explained a large proportion of the variance in individual growth rates, and these family-specific responses to the host were largely consistent cross thermal environments. While most families grew faster on *Veronica* (e.g. family 5), others consistently achieved their highest growth rates on *Plantago* (e.g. family 42). Such intraspecific variation, or in this case variation within the meta-population, for plasticity is common in both natural and laboratory populations (e.g. [20,66]) and hypothesized to be beneficial for insects exposed to climate change [67,68] or inhabiting unpredictable environments [24,69]. We speculate that the spatial variation in host plant abundance in the Åland Islands contributes to the genetic variation for plasticity observed within the Glanville fritillary butterfly metapopulation. When trade-offs in fitness occur across hosts, and adaptation to one host entails a loss of fitness on the other, the slope of the reaction norm represents a genotype-specific level of host specialization. Specialist genotypes are predicted to evolve when host abundance and quality is consistent from generation to generation, even when neither host is inherently better than the other [70,71]. The inconsistent availability of a more suitable host (*Veronica*) in Åland may therefore result in spatially varying selection pressures across the metapopulation, with regions where *Veronica* occurs reliably favouring an increased performance on this host plant, while the evolution of *Plantago* specialists is favoured in patches where *Veronica* is absent.

It is important to place the observed effects of the host plant on larval performance in the context of our experimental design; while temperature interacted with the host plant at the level of the insect herbivore, direct effects of temperature on the host were not assessed. The plants used in the study were cultivated and kept under greenhouse conditions throughout the experiment, and thus represented a high quality diet for the developing larvae. In nature, the two host plants themselves may differ in their responses to variation in temperature, and this interspecific variation may affect the nutritional quality of the available hosts for example in terms of primary nutrients. Additionally, the two host plants used in this study both produce iridoid glycosides [72] (defence chemicals known to deter feeding by generalist insect herbivores [73,74]). Interestingly, these iridoids can also act as feeding stimulants in specialist butterfly larvae [75,76], such as *M. cinxia*, and the concentrations of these secondary compounds are known to be susceptible to variability in precipitation and temperature in *Plantago* [77]. Thus, in the field, when potential effects of temperature on the host plant are present, the interacting effects of temperature and host on insect development demonstrated here could be different.

As a final note we would like to emphasize that the general patterns described in studies like the one presented here (i.e. using a relatively small number of families), may be susceptible to bias when phenotypes vary strongly across genetic backgrounds. For example, using fourteen families we describe that larvae of *M. cinxia* in general perform better on *Veronica* than on *Plantago*, while this is in fact not the case for all families. Therefore, with a different and/or smaller subset of families we could potentially have observed different general patterns.

In conclusion, we demonstrated that larval performance curves in the Glanville fritillary butterfly are family-specific and interactively mediated by the thermal and nutritional environment. The results of our study therefore underscore the importance of studying the multidimensionality of environmental effects on phenotype expression. In addition, our work demonstrates that intraspecific variation is probably an important determinant of population-level responses to environmental change in this system.

Data accessibility. Data available from the Dryad Digital Repository: https://doi.org/10.5061/dryad.kwh70rz2m [78].

Authors' contributions. N.V., S.I., M.S. and E.v.B. conceived and designed the experiments. N.V. performed the experiments. N.V. and E.v.B. analysed the data and wrote the first draft of the manuscript. All authors provided critical feedback and helped to shape the research, analyses and manuscript.

Competing interests. We declare we have no competing interests.

Funding. Financial support was provided by European Research Council (independent starting grant no. 470 637412 'META-STRESS' to M.S.). N.V. was supported by the Erasmus+ programme of the European Union and a Lammi Biological Station's Environmental Research Foundation (LBAYS) grant.

Acknowledgements. We are grateful to Heini Karvinen for practical assistance during the experiments, and Wilco Verberk for suggestions that greatly improved earlier versions of our manuscript. We also thank two anonymous reviewers whose constructive feedback helped to improve our manuscript.

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
