## [Reviewer comments · Proceedings of the Royal Society B: Biological Sciences]

Review History

RSPB-2020-1085.R0 (Original submission)

Review form: Reviewer 1

Recommendation

Reject – article is not of sufficient interest (we will consider a transfer to another journal)

Scientific importance: Is the manuscript an original and important contribution to its field?

Good

General interest: Is the paper of sufficient general interest?

Good

Quality of the paper: Is the overall quality of the paper suitable?

Good

Is the length of the paper justified?

Yes

Should the paper be seen by a specialist statistical reviewer?

No

Do you have any concerns about statistical analyses in this paper? If so, please specify them explicitly in your report.

Yes

It is a condition of publication that authors make their supporting data, code and materials available - either as supplementary material or hosted in an external repository. Please rate, if applicable, the supporting data on the following criteria.

Is it accessible?

Yes

Is it clear?

Yes

Is it adequate?

Yes

Do you have any ethical concerns with this paper?

No

Comments to the Author

The authors examined the interaction between temperature and host plant on fitness-related traits, clutch size, growth rates, fat content and survival, in the Glanville fritillary butterfly. They found that larvae growth rate and fat content depended on the plant and temperature at which they were reared and that the reaction norm for growth rate on different plants varied across the families, suggesting genetic variation in the reaction norm. I liked this paper but one of my concerns was that examined the fitness traits up until the larvae entered diapause and not beyond. I think it would have been worthwhile to assess the impact of low/high-fat content on post-diapausing larvae and adults and to know whether different growth rates and trajectories and long-term fitness impacts. See below for comments on the ms.

Title: I don't really think the authors are examining larval performance curves, reaction norms yes but not curves.

In the first 2 sentences of their abstract they should introduce that they are looking at the interaction between temperature and host plant.

Line 33 I wouldn't refer to host plant as an environmental variable. I guess I always associate with environmental variable with climate, perhaps this is just me but I think it would be better to be more specific.

Line 47 One could argue that avoidance can also occur on the micro-habitat scale as well.

Line 71 Missing in between temperature and the.

Line 75 Consider revising in the light of climate change

Line 84 Is there a host plant preference? Do you tend to find this species more so on one species than the other?

Line 97 remove and

Line 107 What triggers diapause, light or temperature or both?

Line 115 What does it mean that diapausing larvae were stimulated to recommence development? How were they reared in the lab between 2015 and 2019. How many generations between field and lab experiments?

Line 117 Was Plantago the standard plant for culturing the butterflies? Why use Plantago when, as you say on line 96, Veronica reduces mortality?

Line 120 I don't know what the authors mean here by 28:15, where these fluctuating temps and if so can they provide more detail on the fluctuating regime? Is this the standard temperature at which butterflies were reared?

Line 121 I felt these methods were a little confusingly until I looked at their supp figure. I suggest the authors start by simply saying we took eggs from 15 females and split them into two treatments, plant A and B. Second instar larvae were then moved to one of four temperatures

remaining on the same host plant.

Line 134 Is your replication 15 or 14? If it is 14 say so throughout the whole ms.

Line 144 What triggers diapause?

Line 162 These are kind of funny survival curves because you are not really tracking them to they all die, which makes me wonder whether this is really the best way to analyse this data and could perhaps explain the large confidence intervals you get on your curves particular towards the end of survival assessments. What was the proportion of mortality did you get? If I understand your methods correctly you also only have 5-6 data-points for this analysis and maybe only 1 data point where you see significant drop-off and perhaps this is why you see such large error around your estimates.

Line 165 How did you treat your factors in your model, fixed/random?

Line 186 The authors have stated above that survival was not affected by temperature for either plant but was significantly lower on *Plantago* for the two highest temperature. I assuming the authors are referring to the post hoc tests but the should make that apparent. The global model showed this but breaking down the data showed this.

Line 173 What was your replication within families? How many eggs/larvae?

Line 187- 189 These sentences are a little ambiguous and need a little work on their structure. I think all your are trying to say here is that clutch mass tended to increase with increasing temperature. I wonder whether it would actually be better to model this with a regression. i.e. a slope for each temperature per plant and then examine whether the slopes for the different temperatures vary significantly. The fact that mass increases over time isn't really that interesting it is really whether temperature and host plant influences the growth of the clutch. I also couldn't see a temperature x plant interaction in your Figure 2A, it looks as if both plants are doing the same thing with temperature?

Is there a correlation between fat content and growth rate? I am guessing not for *Veronica* but perhaps for *Plantago*. Could be a nice way to think about the mis match between growth rate and fat content, at least in my mind and the very simple view is that you would expect one.

Line 226 The increased variance in *Plantago* is an interesting one. It would be nice to know what optimal growth rates are and whether this optimum shifts depending on host and temperature but to truly get at this you would need to track individuals and measure adult fitness.

Line 239 I am not sure if manipulate is the right word here.

Line 253 I know there can be quite a difference between the top and underside of the leaf and that insects can use plant respiration to evaporatively cool. Have you measured the temperature of leaves and their micro-climates to show that temperatures could get to hot on these plants? I suspect the real issue would have been desiccation stress.

Line 273 Perhaps fat content isn't that important for diapause, what is the length of diapause and is instead important for some other adult trait.

Line 279 It is really only a trade-off if there is a negative effect on fitness, you would need to look at the adults or the post-diapausing larvae mortality rate.

Figure B Interesting that there is no temperature host plant interaction in at least the three you have highlighted. If you grow faster on *Plantago* at the cooler temperature you do so also at warmer temperatures

Review form: Reviewer 2

Recommendation

Accept with minor revision (please list in comments)

Scientific importance: Is the manuscript an original and important contribution to its field?

Good

General interest: Is the paper of sufficient general interest?

Good

Quality of the paper: Is the overall quality of the paper suitable?

Good

Is the length of the paper justified?

Yes

Should the paper be seen by a specialist statistical reviewer?

No

Do you have any concerns about statistical analyses in this paper? If so, please specify them explicitly in your report.

No

It is a condition of publication that authors make their supporting data, code and materials available - either as supplementary material or hosted in an external repository. Please rate, if applicable, the supporting data on the following criteria.

Is it accessible?

N/A

Is it clear?

N/A

Is it adequate?

N/A

Do you have any ethical concerns with this paper?

No

Comments to the Author

Review

Title:

Multidimensional plasticity in the Glanville fritillary butterfly: larval performance curves are temperature, host and family specific.

Summary:

The authors report on an experiment where the effects on larval development of *Melitaea cinxia* of two different host plants was examined at four different temperatures. They found extensive variation in life history traits: growth rate, fat content and survival. There were effects of all variables tested: temperature, diet and family, as well as interactions between the groups and variation within families.

Significance:

This paper highlights the amount of variation in physiological traits that exists just due to changes in the environment (E), and due to different environmental factors interacting with each other (ExE). Additionally, these physiological traits also show extensive variation just due genetic effects (GxExE). Overall, I find the findings of this paper solid, although not exactly surprising. Nonetheless, it is important to identify the different ways in which organisms are affected by their environment, and as such, this is a valuable contribution to field of evolutionary ecology.

I found the study itself thorough and well done. The reporting is straightforward and easy to follow. I have a few major and minor comments that I hope the authors can address.

Major comments:

In line 127, the authors state: "when approximately 90% of the larvae within each group transitioned from the first to the second instar, we generated experimental cohorts of 15 siblings." I'm concerned about this; how many days were there between the first caterpillar reaching 2nd instar vs. approximately 90% reaching that stage? It seems like a wide variation this early on could have major effects on the final calculation of overall growth rate. What did the authors do to ensure this variation very early in development did not affect the reported results?

The authors use a temperature range of 28°C to 34°C. Why were these temperatures chosen? They seem high, especially given that the ambient temperature is much lower. In line 255 the authors allude to the fact that in sunny patches, microhabitats may be much warmer. However, nowhere in the paper is it explicit that the temperature range chosen is representative of actual experienced variation. Could the authors provide a motivation for this temperature range?

Usually, butterflies reared in colder environments grow larger; this is mentioned by the authors in the introduction as well. However, in this paper the opposite is true. I would like to see the authors discuss this discrepancy in their results. Are there other instances where this is the case?

Lastly, what was the maternal diet of the caterpillars in the experiment? Was it a plant-based diet, and if yes, which plant? Could that affect the results? I would like to see more explanation there.

Minor comments:

I personally dislike the term 'early development', it suggests the authors look at the first few days of development. 'Larval development' is a more accurate description.

Line 154; Is there a known effect of sex on relative fat mass?

Line 188; please define 'mean clutch mass'

I find Table S1 very confusing. What do the 1's and 2's mean?

Decision letter (RSPB-2020-1085.R0)

07-Jul-2020

Dear Miss Verspagen:

I am writing to inform you that your manuscript RSPB-2020-1085 entitled "Multidimensional plasticity in the Glanville fritillary butterfly: larval performance curves are temperature, host and family specific." has, in its current form, been rejected for publication in Proceedings B.

This action has been taken on the advice of referees, who have recommended that substantial revisions are necessary. With this in mind we would be happy to consider a resubmission, provided the comments of the referees are fully addressed. However please note that this is not a provisional acceptance.

The resubmission will be treated as a new manuscript. However, we will approach the same reviewers if they are available and it is deemed appropriate to do so by the Editor. Please note

that resubmissions must be submitted within six months of the date of this email. In exceptional circumstances, extensions may be possible if agreed with the Editorial Office. Manuscripts submitted after this date will be automatically rejected.

Sincerely,
Dr Daniel Costa
mailto: proceedingsb@royalsociety.org

Associate Editor
Comments to Author:

Many thanks for submitting your MS, which received thoughtful and detailed comments from two reviewers.

R1 agrees that the subject and your finding is interesting but argues that without understanding how larval behaviour translates into adult fitness, the data do not fully represent fitness.

I agree that not following fitness beyond diapause limits the data's value (and this should be highlighted). However, I recognise the special challenge of getting such data in butterflies. Are there previous studies that show how fat content at diapause affected adult fitness (in males and females?). If not, I think it's important - where post diapause fitness cannot be estimated - to make it explicit that we don't know how these larval traits will affect LRS.

I agree with R2 that the main result of among genotype variation in plasticity across traits, both in relation to temperature and with host plant is important to quantify. And that detailed studies of genetic variation in developmental traits and fitness (and their interaction with the environment) are rare. However, please consider in detail their comments and requests for clarification.

R1 has some concerns with the statistical analysis - especially the effects of non-random mortality (which seem to differ among the families) on variance in your estimates, which I'd like to see you address. Similarly, can you please justify the number of families used, in relation to other studies that have (or have not) shown similar effects.

R2 argues that the study is not surprising, but is valuable. I agree that it is not surprising (i.e. we expect genetic variation in reaction norms, and for them to vary with host and temperature). However, few studies of plasticity quantify these parameters, and often do not consider host plant effects, in the way that is done in this MS.

I also think these data are also important because of how much is known about the ecology of this species, and its interaction with host plants, and this should perhaps be brought out more in the

discussion - i.e. how much the observed host induced variation is likely to affect population demography (and its interaction with temperature) in the field.

I hope careful consideration of the extensive comments on R1 and R2 is useful in improving your manuscript.

Reviewer(s)' Comments to Author:

Referee: 1

Comments to the Author(s)

The authors examined the interaction between temperature and host plant on fitness-related traits, clutch size, growth rates, fat content and survival, in the Glanville fritillary butterfly. They found that larvae growth rate and fat content depended on the plant and temperature at which they were reared and that the reaction norm for growth rate on different plants varied across the families, suggesting genetic variation in the reaction norm. I liked this paper but one of my concerns was that examined the fitness traits up until the larvae entered diapause and not beyond. I think it would have been worthwhile to assess the impact of low/high-fat content on post-diapausing larvae and adults and to know whether different growth rates and trajectories and long-term fitness impacts. See below for comments on the ms.

Title: I don't really think the authors are examining larval performance curves, reaction norms yes but not curves.

In the first 2 sentences of their abstract they should introduce that they are looking at the interaction between temperature and host plant.

Line 33 I wouldn't refer to host plant as an environmental variable. I guess I always associate with environmental variable with climate, perhaps this is just me but I think it would be better to be more specific.

Line 47 One could argue that avoidance can also occur on the micro-habitat scale as well.

Line 71 Missing in between temperature and the.

Line 75 Consider revising in the light of climate change

Line 84 Is there a host plant preference? Do you tend to find this species more so on one species than the other?

Line 97 remove and

Line 107 What triggers diapause, light or temperature or both?

Line 115 What does it mean that diapausing larvae were stimulated to recommence development? How were they reared in the lab between 2015 and 2019. How many generations between field and lab experiments?

Line 117 Was Plantago the standard plant for culturing the butterflies? Why use Plantago when, as you say on line 96, Veronica reduces mortality?

Line 120 I don't know what the authors mean here by 28:15, where these fluctuating temps and if so can they provide more detail on the fluctuating regime? Is this the standard temperature at which butterflies were reared?

Line 121 I felt these methods were a little confusingly until I looked at their supp figure. I suggest the authors start by simply saying we took eggs from 15 females and split them into two treatments, plant A and B. Second instar larvae were then moved to one of four temperatures remaining on the same host plant.

Line 134 Is your replication 15 or 14? If it is 14 say so throughout the whole ms.

Line 144 What triggers diapause?

Line 162 These are kind of funny survival curves because you are not really tracking them to they all die, which makes me wonder whether this is really the best way to analyse this data and could perhaps explain the large confidence intervals you get on your curves particular towards the end of survival assessments. What was the proportion of mortality did you get? If I understand your methods correctly you also only have 5-6 data-points for this analysis and maybe only 1 data point where you see significant drop-off and perhaps this is why you see such large error around your estimates.

Line 165 How did you treat your factors in your model, fixed/random?

Line 186 The authors have stated above that survival was not affected by temperature for either plant but was significantly lower on *Plantago* for the two highest temperature. I assuming the authors are referring to the post hoc tests but the should make that apparent. The global model showed this but breaking down the data showed this.

Line 173 What was your replication within families? How many eggs/larvae?

Line 187- 189 These sentences are a little ambiguous and need a little work on their structure. I think all your are trying to say here is that clutch mass tended to increase with increasing temperature. I wonder whether it would actually be better to model this with a regression. i.e. a slope for each temperature per plant and then examine whether the slopes for the different temperatures vary significantly. The fact that mass increases over time isn't really that interesting it is really whether temperature and host plant influences the growth of the clutch. I also couldn't see a temperature x plant interaction in your Figure 2A, it looks as if both plants are doing the same thing with temperature?

Is there a correlation between fat content and growth rate? I am guessing not for *Veronica* but perhaps for *Plantago*. Could be a nice way to think about the mis match between growth rate and fat content, at least in my mind and the very simple view is that you would expect one.

Line 226 The increased variance in *Plantago* is an interesting one. It would be nice to know what optimal growth rates are and whether this optimum shifts depending on host and temperature but to truly get at this you would need to track individuals and measure adult fitness.

Line 239 I am not sure if manipulate is the right word here.

Line 253 I know there can be quite a difference between the top and underside of the leaf and that insects can use plant respiration to evaporatively cool. Have you measured the temperature of leaves and their micro-climates to show that temperatures could get to hot on these plants? I suspect the real issue would have been desiccation stress.

Line 273 Perhaps fat content isn't that important for diapause, what is the length of diapause and is instead important for some other adult trait.

Line 279 It is really only a trade-off if there is a negative effect on fitness, you would need to look at the adults or the post-diapausing larvae mortality rate.

Figure B Interesting that there is no temperature host plant interaction in at least the three you have highlighted. If you grow faster on *Plantago* at the cooler temperature you do so also at warmer temperatures

Referee: 2

Comments to the Author(s)

Review

Title:

Multidimensional plasticity in the *Glanville* fritillary butterfly: larval performance curves are temperature, host and family specific.

Summary:

The authors report on an experiment where the effects on larval development of *Melitaea cinxia* of two different host plants was examined at four different temperatures. They found extensive variation in life history traits: growth rate, fat content and survival. There were effects of all variables tested: temperature, diet and family, as well as interactions between the groups and variation within families.

Significance:

This paper highlights the amount of variation in physiological traits that exists just due to changes in the environment (E), and due to different environmental factors interacting with each other (ExE). Additionally, these physiological traits also show extensive variation just due genetic effects (GxExE). Overall, I find the findings of this paper solid, although not exactly surprising.

Nonetheless, it is important to identify the different ways in which organisms are affected by their environment, and as such, this is a valuable contribution to field of evolutionary ecology.

I found the study itself thorough and well done. The reporting is straightforward and easy to follow. I have a few major and minor comments that I hope the authors can address.

Major comments:

In line 127, the authors state: “when approximately 90% of the larvae within each group transitioned from the first to the second instar, we generated experimental cohorts of 15 siblings.” I’m concerned about this; how many days were there between the first caterpillar reaching 2nd instar vs. approximately 90% reaching that stage? It seems like a wide variation this early on could have major effects on the final calculation of overall growth rate. What did the authors do to ensure this variation very early in development did not affect the reported results?

The authors use a temperature range of 28°C to 34°C. Why were these temperatures chosen? They seem high, especially given that the ambient temperature is much lower. In line 255 the authors allude to the fact that in sunny patches, microhabitats may be much warmer. However, nowhere in the paper is it explicit that the temperature range chosen is representative of actual experienced variation. Could the authors provide a motivation for this temperature range?

Usually, butterflies reared in colder environments grow larger; this is mentioned by the authors in the introduction as well. However, in this paper the opposite is true. I would like to see the authors discuss this discrepancy in their results. Are there other instances where this is the case?

Lastly, what was the maternal diet of the caterpillars in the experiment? Was it a plant-based diet, and if yes, which plant? Could that affect the results? I would like to see more explanation there.

Minor comments:

I personally dislike the term ‘early development’, it suggests the authors look at the first few days of development. ‘Larval development’ is a more accurate description.

Line 154; Is there a known effect of sex on relative fat mass?

Line 188; please define ‘mean clutch mass’

I find Table S1 very confusing. What do the 1’s and 2’s mean?

Author's Response to Decision Letter for (RSPB-2020-1085.R0)

See Appendix A.

RSPB-2020-2577.R0

Review form: Reviewer 1

Recommendation

Accept as is

Scientific importance: Is the manuscript an original and important contribution to its field?
Good

General interest: Is the paper of sufficient general interest?
Good

Quality of the paper: Is the overall quality of the paper suitable?
Good

Is the length of the paper justified?
Yes

Should the paper be seen by a specialist statistical reviewer?
No

Do you have any concerns about statistical analyses in this paper? If so, please specify them explicitly in your report.
No

It is a condition of publication that authors make their supporting data, code and materials available - either as supplementary material or hosted in an external repository. Please rate, if applicable, the supporting data on the following criteria.

Is it accessible?
Yes

Is it clear?
Yes

Is it adequate?
Yes

Do you have any ethical concerns with this paper?
No

Comments to the Author
I am satisfied with the response to the reviewer's.

Review form: Reviewer 2

Recommendation
Accept as is

Scientific importance: Is the manuscript an original and important contribution to its field?
Excellent

General interest: Is the paper of sufficient general interest?
Excellent

Quality of the paper: Is the overall quality of the paper suitable?
Excellent

Is the length of the paper justified?

Yes

Should the paper be seen by a specialist statistical reviewer?

No

Do you have any concerns about statistical analyses in this paper? If so, please specify them explicitly in your report.

No

It is a condition of publication that authors make their supporting data, code and materials available - either as supplementary material or hosted in an external repository. Please rate, if applicable, the supporting data on the following criteria.

Is it accessible?

N/A

Is it clear?

N/A

Is it adequate?

N/A

Do you have any ethical concerns with this paper?

No

Comments to the Author

The authors have addressed all my concerns. I understand their decision to not further speculate on the size/temperature discrepancy in *M. cinxia*. On a personal note, I think the discussion on diapause strategies is fascinating, and I look forward to forthcoming results. For the paper under I review, I have no further comments, this is great work!

Decision letter (RSPB-2020-2577.R0)

16-Nov-2020

Dear Miss Verspagen

I am pleased to inform you that your manuscript RSPB-2020-2577 entitled "Multidimensional plasticity in the Glanville fritillary butterfly: larval performance is temperature, host and family specific" has been accepted for publication in Proceedings B.

The referee(s) have recommended publication, but also suggest some minor revisions to your manuscript. Therefore, I invite you to respond to the referee(s)' comments and revise your manuscript. Because the schedule for publication is very tight, it is a condition of publication that you submit the revised version of your manuscript within 7 days. If you do not think you will be able to meet this date please let us know.

To revise your manuscript, log into <https://mc.manuscriptcentral.com/prsb> and enter your Author Centre, where you will find your manuscript title listed under "Manuscripts with Decisions." Under "Actions," click on "Create a Revision." Your manuscript number has been appended to denote a revision. You will be unable to make your revisions on the originally

submitted version of the manuscript. Instead, revise your manuscript and upload a new version through your Author Centre.

Sincerely,
Dr Daniel Costa
mailto: proceedingsb@royalsociety.org

Associate Editor
Board Member
Comments to Author:
Both reviewers are happy with the revisions.

I think the additional caveats introduced add to the value of the MS, and I am happy to accept the authors' request to not speculate (here) as to why this species deviates from the usual body size/temperature relationship - given the possible causes of it being an exception is potentially an interesting part of the species life history, and something that is under investigation.

I would be keen to see the results of the survival curves GLM included in the SI, as suggested in the response to reviewer 1.

I would also be keen for the ability of individuals to limit their exposure to temperature variation in the field via microclimate selection (provided such microclimate remain available) to be mentioned in the text. I think this is a critical (and sometimes overlooked) way that exposure is minimised in the field (or, sometimes, increased via maladaptive behaviour in novel conditions). I hope this doesn't disrupt the flow of the text too much!

Reviewer(s)' Comments to Author:
Referee: 1

Comments to the Author(s).
I am satisfied with the response to the reviewer's.

Referee: 2

Comments to the Author(s).
The authors have addressed all my concerns. I understand their decision to not further speculate on the size/temperature discrepancy in *M. cinxia*. On a personal note, I think the discussion on diapause strategies is fascinating, and I look forward to forthcoming results. For the paper under I review, I have no further comments, this is great work!

Author's Response to Decision Letter for (RSPB-2020-2577.R0)

See Appendix B.

Decision letter (RSPB-2020-2577.R1)

23-Nov-2020

Dear Miss Verspagen

I am pleased to inform you that your manuscript entitled "Multidimensional plasticity in the Glanville fritillary butterfly: larval performance is temperature, host and family specific" has been accepted for publication in Proceedings B.

Open Access

You are invited to opt for Open Access, making your freely available to all as soon as it is ready for publication under a CCBY licence. Our article processing charge for Open Access is £1700. Corresponding authors from member institutions (<http://royalsocietypublishing.org/site/librarians/allmembers.xhtml>) receive a 25% discount to these charges. For more information please visit <http://royalsocietypublishing.org/open-access>.

Paper charges

Sincerely,
Editor, Proceedings B
<mailto:proceedingsb@royalsociety.org>

Appendix A

Helsinki, October 15th, 2020

Dear Prof. Daniel Costa,

We would like to thank you for the opportunity to resubmit a revised version of our manuscript, now entitled “*Multidimensional plasticity in the Glanville fritillary butterfly: larval performance is temperature, host and family specific*”. The previous reference number is ID RSPB-2020-1085. We are grateful to the reviewers for providing constructive comments on the previous version of the manuscript. Many of the suggestions were extremely useful and greatly strengthened our manuscript.

We have addressed all these major comments and changed the manuscript accordingly. In the point-by-point response provided below, we reply to all comments in **blue font**. When appropriate, these responses correspond to the highlighted sections in the revised manuscript, again using **blue font**. We have also taken this opportunity to acknowledge reviewers in the Acknowledgement section of our paper. We believe that the revised version of our manuscript will be of interest to the readership of the *Proceedings B* having gone through this process of revision. We hope that reviewers, and editorial team alike, will agree.

Kind regards,

Nadja Verspagen (on the behalf of all authors)

Associate Editor:

Many thanks for submitting your MS, which received thoughtful and detailed comments from two reviewers.

R1 agrees that the subject and your finding is interesting but argues that without understanding how larval behaviour translates into adult fitness, the data do not fully represent fitness.

I agree that not following fitness beyond diapause limits the data's value (and this should be highlighted). However, I recognise the special challenge of getting such data in butterflies. Are there previous studies that show how fat content at diapause affected adult fitness (in males and females?). If not, I think it's important - where post diapause fitness cannot be estimated - to make it explicit that we don't know how these larval traits will affect LRS.

I agree with R2 that the main result of among genotype variation in plasticity across traits, both in relation to temperature and with host plant is important to quantify. And that detailed studies of genetic variation in developmental traits and fitness (and their interaction with the environment) are rare. However, please consider in detail their comments and requests for clarification.

R1 has some concerns with the statistical analysis - especially the effects of non-random mortality (which seem to differ among the families) on variance in your estimates, which I'd like to see you address. Similarly, can you please justify the number of families used, in relation to other studies that have (or have not) shown similar effects.

R2 argues that the study is not surprising, but is valuable. I agree that it is not surprising (i.e. we expect genetic variation in reaction norms, and for them to vary with host and temperature). However, few studies of plasticity quantify these parameters, and often do not consider host plant effects, in the way that is done in this MS. I also think these data are also important because of how much is known about the ecology of this species, and its interaction with host plants, and this should perhaps be brought out more in the discussion - i.e. how much the observed host induced variation is likely to affect population demography (and its interaction with temperature) in the field.

I hope careful consideration of the extensive comments on R1 and R2 is useful in improving your manuscript.

We are delighted with the positive and constructive feedback from the associate editor and the reviewers, and were pleased to read that they found our findings interesting and important, especially in the context of the well-known ecology of our model system. Most of the comments summarized by the Associate Editor are addressed in the point-by-point response to the reviewers. The remaining two comments from the AE are addressed below.

1) Can you please justify the number of families used, in relation to other studies that have (or have not) shown similar effects.

*Laboratory studies using *M. cinxia* are logistically challenging due to their gregarious lifestyle and obligatory diapausing period, and for logistic reasons the average sample size in studies like ours is typically between 1500 and 2000 individuals (N=1680 in our study). The number of families used in our work is somewhat lower than those conducted in the first decade of this century (e.g. van Nouhuys, Singer, & Nieminen, 2003; Saastamoinen, van Nouhuys, Nieminen, O'Hara, & Suomi, 2007). Since these studies solely aimed to shed light on the effect of one treatment, the host plant, on pre-diapause performance, omitting potential family-specific responses and interactions with the thermal environment, these researches were able to include a large number of families. Since we aimed to explore the relative contributions of different sources of phenotypic variance, the number of families had to be reduced to facilitate the controlled inclusion of additional environmental variation. With a smaller number of families we find, with regards to the effect of the host, similar effects as described in previous studies. However, given our emphasis on family-specific responses we are able to highlight the importance of intraspecific variation for this trait. In the penultimate paragraph we discuss that these general patterns may be susceptible to bias when phenotypes vary strongly across genetic backgrounds and a relatively small number of families is used.*

2) I also think these data are important because of how much is known about the ecology of this species, and its interaction with host plants, and this should perhaps be brought out more in the discussion

*We agree that our results are particularly interesting in the context of the detailed understanding of population dynamics in this ecological system. We have now added more details to the discussion about how the results of our laboratory study relate to the conditions that *M. cinxia* faces in the wild.*

Referee 1:

The authors examined the interaction between temperature and host plant on fitness-related traits, clutch size, growth rates, fat content and survival, in the Glanville fritillary butterfly. They found that larvae growth rate and fat content depended on the plant and temperature at which they were reared and that the reaction norm for growth rate on different plants varied across the families, suggesting genetic variation in the reaction norm. I liked this paper but one of my concerns was that examined the fitness traits up until the larvae entered diapause and not beyond. I think it would have been worthwhile to assess the impact of low/high-fat content on post-diapausing larvae and adults and to know whether different growth rates and trajectories and long-term fitness impacts.

First of all, we would like to thank the reviewer for their comprehensive work. We were glad to read that they enjoyed reading our manuscript. We agree that it would have been an interesting addition to our manuscript if we would have been able to track larvae after diapause to assess the effect of our treatments on lifetime reproductive success. This was unfortunately not possible or feasible for several reasons. Firstly, larvae had to be sacrificed to perform the fat content analyses, and it was therefore no longer possible to track these individuals beyond diapause. In addition, we are unable to track individual larvae throughout development (with repeated trait assessments) due to the gregarious nature of the larvae and the inability to mark them at this stage. In other words, an individual with a fast pre-diapause growth rate has to be kept with other unmarked conspecifics during diapause, making it impossible to assess how this fast growth rate correlates with

other life-history traits later in life at the individual level. Finally, the species takes nearly a year to complete its life cycle in the laboratory, including a five month obligatory diapause period, making studies that address the effects of early life experiences on reproductive success particularly challenging and not possible in the light of the current MSc study. We added a statement to the revised manuscript to make the reader aware that this limitation could affect the interpretation of our results (lines 250-256).

See below for comments on the ms.

Title: I don't really think the authors are examining larval performance curves, reaction norms yes but not curves.

We have now changed the title to: Multidimensional plasticity in the Glanville fritillary butterfly: larval performance is temperature, host and family specific.

In the first 2 sentences of their abstract they should introduce that they are looking at the interaction between temperature and host plant.

*We have rephrased the second sentence of the abstract which now reads as follows (lines 25-27):
'Here, we study how variation in temperature and host plant, i.e. the consequences of potential maternal oviposition choices, affects a suite of life-history traits in pre-diapause larvae of the Glanville fritillary butterfly'.*

Line 33 I wouldn't refer to host plant as an environmental variable. I guess I always associate with environmental variable with climate, perhaps this is just me but I think it would be better to be more specific.

This has been changed in the revised version of the manuscript (line 33).

Line 47 One could argue that avoidance can also occur on the micro-habitat scale as well.

Absolutely! Avoidance could also occur on a more micro-habitat scale but to not interfere with the flow of the text too much we prefer to only give one example of how organisms may avoid environmental stress (...e.g. through tracking favourable conditions by expanding to higher latitudes or altitudes).

Line 71 Missing in between temperature and the.

Thanks for spotting this! We have added the missing word (line 68).

Line 75 Consider revising in the light of climate change

This has been revised (line 71).

Line 84 Is there a host plant preference? Do you tend to find this species more so on one species than the other?

This is a very interesting questions with a somewhat complicated answer since host-use (and likely host preference) in this species varies on a spatial scale across the butterflies range as well as within the metapopulation. We describe some of the spatial patterns in detail in the discussion of this manuscript (lines 302-311, no change). In addition, ongoing work in our research group has revealed strong temporal variation in host use in the Glanville fritillary metapopulation, and we are able to show that this variation is strongly associated to the amount of precipitation within the habitat patches. Prolonged periods of drought, especially in May or July, result in a relative bias towards the use of Plantago.

Line 97 remove and

We believe the use of 'and' in this sentence is correct.

Line 107 What triggers diapause, light or temperature or both?

*In contrast to many multivoltine insects, diapause in this univoltine butterfly species is obligatory, meaning that they will even diapause when conditions for survival and continuing development would remain optimal (in regards to both temperature and light). The role of environmental factors on diapause and/or diapause strategy, such as photoperiod or temperature triggering diapause in species with facultative diapause, are therefore not well understood in *M. cinxia*. Current research efforts in our group are directed towards shedding more light on the role of environmental factors in shaping diapause strategies in this species.*

We have added more information about the diapause period to the manuscript, including a notion that diapause is obligatory in this species (lines 112-113).

Line 115 What does it mean that diapausing larvae were stimulated to recommence development? How were they reared in the lab between 2015 and 2019. How many generations between field and lab experiments?

*Laboratory populations spend up to six months as diapausing larvae in a climate-controlled chamber set to 5 °C, mimicking the winter conditions in the wild. The parental generation of the individuals used in this study were removed from this cooling chamber after which the increase in ambient temperature, light and moisture stimulates them to recommence feeding and thus development (i.e. become post-diapause larvae who then complete two more instars before metamorphosis). Laboratory populations of *M. cinxia* larvae are typically reared on *Plantago* (more information below) in climate-controlled cabinets set to 28 °C during the day and 15 °C during the night. These temperatures reflect ambient temperature (and daily fluctuations) during summer as measured during the day and night close to the ground where the larvae develop (see lines 268-275 in the revised manuscript). Since this species can only complete one generation per year and the stock was collected as diapausing larvae in 2015 this (actively outcrossed) population completed four generations in the laboratory before the experiments were conducted.*

We have added more information about the species life cycle and laboratory rearing to the manuscript (lines 108-113, 115-119).

Line 117 Was *Plantago* the standard plant for culturing the butterflies? Why use *Plantago* when, as you say on line 96, *Veronica* reduces mortality?

Plantago is indeed the standard plant used to maintain laboratory stocks of M. cinxia. This plant species is present in each of the natural habitats of the species in Åland, while this is not the case for Veronica (see lines 302-311). In addition, this host is a bit easier to cultivate and it produces slightly more biomass per plant than Veronica. Finally, Plantago is also used to keep laboratory protocols consistent over time.

We have added more information about laboratory rearing to the manuscript (lines 108-113).

Line 120 I don't know what the authors mean here by 28:15, where these fluctuating temps and if so can they provide more detail on the fluctuating regime? Is this the standard temperature at which butterflies were reared?

As mentioned above, laboratory populations of M. cinxia larvae are typically reared on Plantago in climate-controlled cabinets set to 28 °C during the day and 15 °C during the night. These temperature reflect ambient temperature (and daily fluctuations) during summer as measured during the day and night close to the ground in open spaces where the larvae develop (see lines 138-140 and 268-275 in the revised manuscript).

We have added more information about laboratory rearing to the manuscript (lines 130-132).

Line 121 I felt these methods were a little confusingly until I looked at their supp figure. I suggest the authors start by simply saying we took eggs from 15 females and split them into two treatments, plant A and B. Second instar larvae were then moved to one of four temperatures remaining on the same host plant.

We agree that these methods were written down in a confusing manner and we have implemented the suggested changes (see lines 123-134 of the revised manuscript).

Line 134 Is your replication 15 or 14? If it is 14 say so throughout the whole ms.

Indeed, replication at the level of family in our manuscript is 14. Our objective was to include full-sibs of 15 families but unfortunately one female did not produce enough offspring to complete all treatments, and these data were excluded from further analyses. We mentioned this in the previous version of the manuscript to be candid and upfront, but now realize that this sincerity is likely to confuse the reader. We have now removed all references to female '4' from the manuscript.

Line 144 What triggers diapause?

Please see comment above.

Line 162 These are kind of funny survival curves because you are not really tracking them to they all die, which

makes me wonder whether this is really the best way to analyse this data and could perhaps explain the large confidence intervals you get on your curves particular towards the end of survival assessments. What was the proportion of mortality did you get? If I understand your methods correctly you also only have 5-6 data-points for this analysis and maybe only 1 data point where you see significant drop-off and perhaps this is why you see such large error around your estimates.

The reviewer is right in that this is not a traditional survival analysis where we track individuals throughout their lives, instead we track them until they enter diapause or die. Larvae were censored at the time point of entering diapause (since they are no longer 'at risk') and as a consequence, as time progresses sample size is decreasing which explains the large confidence intervals towards the end of survival assessments. We believe that this is interesting in its own right, since it reflects that the slowly developing individuals are the ones that do not survive the pre-diapause stage. This is also why we decided to represent distribution of time-to-diapause above the survival curves, demonstrating that mortality risk increases significantly after the optimal (or mean) moment for diapause has been reached. To confirm our key result we performed an additional analysis using only the end-point data (i.e. whether larvae survived until diapause or not). For this, we used a generalized linear model (GLM) with family, temperature and host plant as fixed effects. Tukey HSD tests confirmed the results of our Kaplan-Meier approach with significant differences in the likelihood of survival between the two host plants only occurring at the two highest temperatures. This additional analysis confirmed that the results as presented in the manuscript are robust and we believe that the Kaplan-Meier approach, which in addition shows that mortality mainly occurs after the optimal day of diapause is more informative. As a final note regarding sample sizes, for each of the 8 treatments we have 14 families with 15 individuals (at time = 0) and all these larvae were tracked until they entered diapause or until time = 24 (last check point).

Given that these are indeed 'kind of funny survival curves' we have made changes to the manuscript to clarify our results and the representation of these (see lines 166-167 and 581-583). We have also added the survival proportion (line 185). We would consider adding the results from the GLM to the supplementary material upon request from the reviewers or editorial board.

Line 165 How did you treat your factors in your model, fixed/random?

All variables were included as fixed factors in the full models, and we have made changes to the manuscript to specify this (lines 172 and 173).

Line 186 The authors have stated above that survival was not affected by temperature for either plant but was significantly lower on *Plantago* for the two highest temperature. I assuming the authors are referring to the post hoc tests but the should make that apparent. The global model showed this but breaking down the data showed this.

Because of the type of model used, it was not possible to perform traditional post hoc tests. Instead, we tested the influence of temperature within each host plant, and vice versa. We have now clarified this in the revised manuscript (187-191)

Line 173 What was your replication within families? How many eggs/larvae?

We used 15 individuals per treatment (host plant [x2], temperature [x4] and family [x14]), yielding 1680 larvae in total. We have made changes to the manuscript to make the sample sizes used more clear (lines 134, 158 and 185).

Line 187- 189 These sentences are a little ambiguous and need a little work on their structure. I think all you are trying to say here is that clutch mass tended to increase with increasing temperature. I wonder whether it would actually be better to model this with a regression. i.e. a slope for each temperature per plant and then examine whether the slopes for the different temperatures vary significantly. The fact that mass increases over time isn't really that interesting it is really whether temperature and host plant influences the growth of the clutch. I also couldn't see a temperature x plant interaction in your Figure 2A, it looks as if both plants are doing the same thing with temperature?

There are a variety of models available to analyse these type of data, in our case, the interaction of the environmental factors with time (i.e. day:Temperature and day:HostPlant) represent variation in slopes. day:Temperature interaction signifies that slopes are steeper at higher temperatures, which we agree is not really surprising. But the day:HostPlant interaction signifies that slopes are steeper when larvae utilize Veronica (i.e. mass gain over time is faster on this host). We have improved the wording of this section to make the results clearer (lines 192-196).

Is there a correlation between fat content and growth rate? I am guessing not for Veronica but perhaps for Plantago. Could be a nice way to think about the mismatch between growth rate and fat content, at least in my mind and the very simple view is that you would expect one.

We would actually predict the correlation between fat content and growth rate to be temperature and host-specific, and potentially even be dependent on the genetic background (i.e. families). Upon recommendation of the reviewer we have now looked into the correlations between these two variables but did not find a clear pattern within the treatments. A trend may appear when groups are analysed in combination, but disappear (or reverse) when groups are looked at independently (Simpson's paradox), making it hard to identify true relationships.

Line 226 The increased variance in Plantago is an interesting one. It would be nice to know what optimal growth rates are and whether this optimum shifts depending on host and temperature but to truly get at this you would need to track individuals and measure adult fitness.

This is indeed very interesting and we agree that it would have been nice to track individuals and measure adult fitness. Sadly this was not possible in this study (see previous comment).

Line 239 I am not sure if manipulate is the right word here.

We have replaced the word manipulate with change (line 241)

Line 253 I know there can be quite a difference between the top and underside of the leaf and that insects can use plant respiration to evaporatively cool. Have you measured the temperature of leaves and their microclimates to show that temperatures could get to hot on these plants? I suspect the real issue would have been desiccation stress.

Indeed, there can be differences in temperature in different parts on the leaf. We have not measured these, but unpublished data collected by other researchers in our group confirm that the temperature of post-diapause larvae indeed corresponds with the thermal stratification mentioned in the manuscript (lines 258-263).

Line 273 Perhaps fat content isn't that important for diapause, what is the length of diapause and is instead important for some other adult trait.

It may indeed be the case that fat accumulated during larval development becomes important during later life stages. However, because we were unable to test this hypothesis we prefer to not speculate about this topic in the manuscript.

Line 279 It is really only a trade-off if there is a negative effect on fitness, you would need to look at the adults or the post-diapausing larvae mortality rate.

We replaced the word "trade-off" with "compromise" (see line 286)

Figure B Interesting that there is no temperature host plant interaction in at least the three you have highlighted. If you grow faster on Plantago at the cooler temperature you do so also at warmer temperatures

Thanks for this! We agree that this is a very interesting result!

Referee 2:

Title:

Multidimensional plasticity in the Glanville fritillary butterfly: larval performance curves are temperature, host and family specific.

Summary:

The authors report on an experiment where the effects on larval development of *Melitaea cinxia* of two different host plants was examined at four different temperatures. They found extensive variation in life history traits: growth rate, fat content and survival. There were effects of all variables tested: temperature, diet and family, as well as interactions between the groups and variation within families.

Significance:

This paper highlights the amount of variation in physiological traits that exists just due to changes in the environment (E), and due to different environmental factors interacting with each other (ExE). Additionally,

these physiological traits also show extensive variation just due genetic effects (GxExE). Overall, I find the findings of this paper solid, although not exactly surprising. Nonetheless, it is important to identify the different ways in which organisms are affected by their environment, and as such, this is a valuable contribution to field of evolutionary ecology.

I found the study itself thorough and well done. The reporting is straightforward and easy to follow. I have a few major and minor comments that I hope the authors can address.

We would like to thank the reviewer for the positive response regarding our work!

Major comments:

In line 127, the authors state: “when approximately 90% of the larvae within each group transitioned from the first to the second instar, we generated experimental cohorts of 15 siblings.” I’m concerned about this; how many days were there between the first caterpillar reaching 2nd instar vs. approximately 90% reaching that stage? It seems like a wide variation this early on could have major effects on the final calculation of overall growth rate. What did the authors do to ensure this variation very early in development did not affect the reported results?

This is a valid concern but we would like to clarify that the vast majority of larvae within a cohort transition to the next instar on the same day (i.e., if we found a couple of larvae in the second instar during the morning check, 90% of the larvae would have transitioned by the early afternoon). Transitions are particularly synchronized during early development. The 90% threshold was applied to ensure that individuals making up the tail of the distribution did not significantly affect our data. So to give a direct answer to the question from the reviewer: the number of days between the first caterpillar reaching 2nd instar vs. approximately 90% reaching that stage was typically 0 and never more than 1. In the revised version of the methods section we aimed to make this clearer (lines 126-128).

The authors use a temperature range of 28°C to 34°C. Why were these temperatures chosen? They seem high, especially given that the ambient temperature is much lower. In line 255 the authors allude to the fact that in sunny patches, micro-habitats may be much warmer. However, nowhere in the paper is it explicit that the temperature range chosen is representative of actual experienced variation. Could the authors provide a motivation for this temperature range?

*Laboratory populations of *M. cinxia* larvae are typically reared on *Plantago* (see also above) in climate-controlled cabinets set to 28 °C during the day and 15 °C during the night. Though these temperatures seem high, they reflect ambient temperature (and daily fluctuations) during summer as measured during the day and night close to the ground in open spaces where the larvae develop (see also lines 259-263 in the revised manuscript). In this manuscript we aimed to explore the effect of hotter and/or more sunny micro-climates on larval development. Starting from the standard protocol (28:15 °C) we increased the day temperatures in steps of 2 °C. Note that the temperatures during the night remained at 15 °C (i.e. no sunlight) and therefore the mean temperature change between treatments only occurs in increments of 1 °C. The highest temperature used here*

(34°C during the day, reflecting ambient temperatures of 14-22 °C with thermal stratification) does not exceed the upper limit of the normal temperature range observed during the summer on Åland (see figure S5). We have made changes to the manuscript to justify the temperature range chosen (lines 130-134).

Usually, butterflies reared in colder environments grow larger; this is mentioned by the authors in the introduction as well. However, in this paper the opposite is true. I would like to see the authors discuss this discrepancy in their results. Are there other instances where this is the case?

*The negative association between temperature and body size in this species has been observed and described before (e.g. Kallioniemi & Hanski, 2011; Saastamoinen, Ikonen, Wong, Lehtonen, & Hanski, 2013), and our results are consistent with these earlier studies. This discrepancy has been discussed in these reports and was hypothesized to be associated with the fact that this study population occurs at the northern range limit of the species and thermal conditions may be a limiting factor. However, ongoing work in our research group, involving detailed assessments of the number of instars each larvae completes before diapause, suggests that environmental factors (such as temperature and photo-period) may affect the diapausing strategies employed by this species (Kahilainen et al. in preparation). The comment from the reviewer made us realize that these environmentally induced strategies may provide an alternative hypothesis for the fact that the temperature-size rule does not seem to apply to *M. cinxia*. If the number of pre-diapause instars are indeed affected by environmental factors (results of laboratory studies pending) the large-bodied individuals could be those that opted to include an additional instar before entering diapause. Unfortunately, such analyses of diapausing strategies involve either meticulous tracking of larval development or detailed assessments of head capsules, neither were done in this study. Since our study mainly focusses on three larval traits; early survival, early growth rates and relative fat content at diapause (with body size and development time in the supplement), and because research on the potential role of environmental factors on diapausing strategies is ongoing, we prefer not to speculate about the species-specific deviation from temperature-size rule at this stage. We hope that the reviewers and the editorial team are willing accept this decision.*

Lastly, what was the maternal diet of the caterpillars in the experiment? Was it a plant-based diet, and if yes, which plant? Could that affect the results? I would like to see more explanation there.

*Laboratory populations of *M. cinxia* larvae are typically reared on *Plantago*, and this was also true for the parental generation that produced the experimental larvae. In theory the maternal diet could affect our results since transgenerational effects are widespread and have the potential to be strong and persistent (Yin, Zhou, Lin, Li & Zhang, 2019 but see also Sánchez-Tójar et al., 2020). These potential effects of the maternal diet have not been researched in this species, a gap in knowledge that we aim to fill in the near future. Given that we lack a complete (or even a partial) understanding of the impact of the maternal host plant on LH traits of her offspring, we believe that a detailed discussion of this topic in this manuscript would be speculative and uninformative. To incorporate the questions from the reviewer regarding the diet of the parents we have made changes clarify that this generation was also reared using a plant-based diet (*Plantago*) (lines 117-118).*

Minor comments:

I personally dislike the term 'early development', it suggests the authors look at the first few days of development. 'Larval development' is a more accurate description.

We agree and have changed this throughout the manuscript.

Line 154; Is there a known effect of sex on relative fat mass?

*Our study is the first to quantify relative fat content in this species. Unfortunately, we were not able to explore potential sex-specific effects since the sex of the individual cannot be determined at the pre-diapause stage. The relative fat content of adults has been quantified in other Lepidoptera. For example, in *Mycalesina* butterflies the relative fat content of males is greater than of females (van Bergen et al., 2017), and male mating success is positively associated with their relative fat content (Kehl et al., 2015). Males of *Bicyclus anynana* also have a higher fat percentage than females. Nutritional limitation reduces fat percentage in both sexes whereas impacts of developmental thermal conditions are sex-dependent (Saastamoinen, Brommer, Brakefield, & Zwaan, 2013)*

Line 188; please define 'mean clutch mass'

The 'mean clutch mass' is indeed not an accurate description of this trait, and we thank the reviewer for pointing this out. This value in fact represents 'mean larval mass' since it was obtained by dividing the total clutch mass by the remaining number of individuals in the clutch (to account for variation in survival across treatments). We have added this description to the methods (lines 145-147) and have made changes to the results section accordingly.

I find Table S1 very confusing. What do the 1's and 2's mean?

Our full-factorial experimental design yielded 8 treatments with 15 siblings each, hence we required at least 120 offspring from each female. For five females, offspring from a second egg clutch, in addition to the first one, had to be used to complete all experimental treatments (see lines 136-137). The 1's and 2's in Table S1 indicate whether the individuals in that treatment were derived from the females' first or second clutch. We have now made changes to the table description to clarify this. In addition, upon recommendation of R1 we have removed the information from one female who did not produce enough offspring to complete all treatments and data from this family had (already) been excluded from further analyses.

Literature

- van Bergen, E., Osbaldeston, D., Kodandaramaiah, U., Brattström, O., Aduse-Poku, K., & Brakefield, P. M. (2017). Conserved patterns of integrated developmental plasticity in a group of polyphenic tropical butterflies. *BMC Evolutionary Biology*, 17(1), 59. doi:10.1186/s12862-017-0907-1
- Kallioniemi, E., & Hanski, I. (2011). Interactive effects of Pgi genotype and temperature on larval growth and survival in the Glanville fritillary butterfly. *Functional Ecology*, 25(5), 1032-1039.
- Kehl, T., Bensch, J., Böhm, F., Kniepkamp, B. O., Leonhardt, V., Schwieger, S., & Fischer, K. (2015). Fat and sassy: factors underlying male mating success in a butterfly. *Entomologia Experimentalis Et Applicata*, 155(3), 257-265. doi:10.1111/eea.12305
- van Nouhuys, S., Singer, M. C., & Nieminen, M. (2003). Spatial and temporal patterns of caterpillar performance and the suitability of two host plant species. *Ecological Entomology*, 28(2), 193-202. doi:10.1046/j.1365-2311.2003.00501.x
- Saastamoinen, M., Brommer, J. E., Brakefield, P. M., & Zwaan, B. J. (2013). Quantitative genetic analysis of responses to larval food limitation in a polyphenic butterfly indicates environment- and trait-specific effects. *Ecology and Evolution*, 3(10), 3576-3589. doi:10.1002/ece3.718
- Saastamoinen, M., Ikonen, S., Wong, S. C., Lehtonen, R., & Hanski, I. (2013). Plastic larval development in a butterfly has complex environmental and genetic causes and consequences for population dynamics. *Journal of Animal Ecology*, 82(3), 529-539. doi: <https://doi.org/10.1111/1365-2656.12034>
- Saastamoinen, M., van Nouhuys, S., Nieminen, M., O'Hara, B., & Suomi, J. (2007). Development and survival of a specialist herbivore, *Melitaea cinxia*, on host plants producing high and low concentrations of iridoid glycosides. Paper presented at the Annales Zoologici Fennici.
- Sánchez-Tójar, A., Lagisz, M., Moran, N. P., Nakagawa, S., Noble, D. W. A., & Reinhold, K. (2020). The jury is still out regarding the generality of adaptive 'transgenerational' effects. *Ecology Letters*. doi:10.1111/ele.13479
- Yin, J., Zhou, M., Lin, Z., Li, Q. Q., & Zhang, Y.-Y. (2019). Transgenerational effects benefit offspring across diverse environments: a meta-analysis in plants and animals. *Ecology Letters*, 22(11), 1976-1986. doi:10.1111/ele.13373

Appendix B

Helsinki, 19th November 2020

Dear Prof. Daniel Costa,

We would like to thank you for the acceptance of our manuscript titled “*Multidimensional plasticity in the Glanville fritillary butterfly: larval performance is temperature, host and family specific*”. We have addressed the final suggestions by the Associate Editor, namely to include the additional survival GLMM in the supplementary materials and to mention the ability of individuals to respond to unfavourable conditions by selecting more suitable microclimates in the manuscript. A version of the manuscript and supplementary materials with tracked changes (in **blue font**) can be found below.

Kind regards,

Nadja Verspagen (on the behalf of all authors)